# Early diagnosis of skin cancer from phone-taken skin lesion images using Vision Transformers

**Sina Garazhian**[*1] 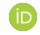          SINA.GARAZHIAN@MPINAT.MPG.DE

[1] *Research group of Quantitative and Systems Biology, Max-Planck-Institute for Multidisciplinary Sciences (MPI-NAT), University of Göttingen, Göttingen, Germany*

**Parsa Hariri**[*2]

HARIRI.PARSA@HELMHOTLZ-MUNICH.DE

[2] *Computational Discovery Research Group, Institute of Computational Biology, Helmholtz Zentrum München – German Research Center for Environmental Health (GmbH), Neuherberg, Germany*
*Department of Life Sciences, Technical University of Munich (TUM), Munich, Germany*

**Editors:** Under Review for MIDL 2025

## Abstract

Recent advances in computer vision have made Vision Transformers (ViTs) strong alternatives to CNNs in medical imaging. We compare top ViT models—including Token-to-Token ViT, CaiT, LeViT, ATSViT, and XCiT—on the Kaggle skin cancer dataset, focusing on classification accuracy, real score, and model complexity. While ViTs for small datasets show high accuracy, they have many parameters; LeViT offers strong performance with the fewest parameters. This review highlights current trends, deployment challenges, and future directions for transformers in skin cancer detection.

**Keywords:** Vision Transformers, Deep Learning, Skin Cancer, Dermatology images.

## 1. Introduction

Transformers, first designed for NLP tasks, gained popularity through models like BERT and RoBERTa (Vaswani et al., 2017; Devlin et al., 2019; Liu et al., 2019). Their success led to applications in computer vision (CV), where CNNs had traditionally dominated (He et al., 2016a,b; Tan and Le, 2019). Early ViTs combined attention with convolution (Bello et al., 2019), but newer versions rely solely on self-attention.

ViTs have since been applied to image classification (Dosovitskiy et al., 2020; Touvron et al., 2021a), segmentation (Ye et al., 2019a), object detection (Ye et al., 2019b), and video analysis (Sun et al., 2019). The original Vision Transformer (Dosovitskiy et al., 2020) showed pure transformer models could excel in CV, inspiring further research. Studies (Azad et al., 2024; Liu et al., 2023) explored ViTs in medical imaging, and (Khalil et al., 2023) reviewed their evolution into lighter, efficient models.

While most reviews focus on clinic-acquired images from mid-to-late cancer stages, our review evaluates recent ViTs for early skin cancer detection using phone-quality lesion images.

---

[*] Contributed equally

## 2. Dataset

We used the ISIC 2024 Kaggle dataset (Kurtansky et al., 2024), which includes 401K 3D Total Body Photography (TBP) images mimicking non-dermoscopic photos. Captured with the Vectra WB360, the images cover the full skin surface. AI software detects and crops individual lesions into 15×15 mm images.

## 3. Results

Our results in Figure 1 highlight a non-linear relationship between model complexity and real-world performance. The Vision Transformer for Small Datasets (Lee et al., 2021) achieved the highest real score of 132, indicating superior performance and generalization by incorporating Shifted Patch Tokenization (SPT) and Locality Self-Attention (LSA) which increases the receptive field during tokenization and sharpens attention scores, respectively. Notably, it did so with a moderate parameter count (∼54M) and 50 training epochs, showcasing that well-designed, domain-adapted ViTs can outperform larger architectures when carefully tuned for small-scale medical datasets. Despite having the same accuracy (92%) as several other models, its higher real score suggests better optimization and convergence behavior over training.

In contrast, CaiT (Touvron et al., 2021b) which utilizes LayerScale, a learnable per-channel residual scaling mechanism that facilitates the training of deep transformers, the largest model in our study with over 120 million parameters, underperformed significantly with a real score of 97, the lowest among all models including the plain ViT model (Beyer et al., 2022). Although it reached a marginally higher accuracy (93%), its short training duration (10 epochs) likely hindered its potential. This illustrates the importance of not only model capacity but also sufficient training time for transformers to fully utilize their representational power.

LeViT (Graham et al., 2021) which employs a multi-stage transformer design incorporating CNN-like components, stands out as the most efficient model in our benchmark, achieving a real score of 125 and the highest classification accuracy (94%) with a remarkably small footprint of just 17M parameters. This model is particularly well-suited for real-time or embedded diagnostic applications, where computational resources are limited. The result also confirms the efficacy of hybrid convolution-attention designs in achieving competitive performance with minimal complexity. Token-to-Token ViT (Yuan et al., 2021) which improves the tokenization process by recursively aggregating neighboring tokens, preserving local structure through a Tokens-to-Token transformation, and ats ViT (Fayyaz et al., 2022) which introduces a parameter-free Adaptive Token Sampler (ATS) module that dynamically selects informative tokens per input image. This adaptivity allows for significant reduction in token count and Giga Floating Point Operations per second (GFLOPs) during inference, each scoring 113 in real score with comparable accuracies (92%), illustrate the potential of patch re-encoding and attention-based scaling for performance gains. However, the Token-to-Token ViT model encountered memory issues during training, emphasizing a practical limitation despite its otherwise balanced architecture and low parameter count (∼20M).

Emerging architectural innovations such as patch merging and cross-covariance attention also showed strong results. The xcit (El-Nouby et al., 2021) model which proposes a novel Cross-Covariance Attention (XCA) mechanism that operates across feature channels instead

of tokens with real score of 122, $\sim$12M parameters and vit_with_patch_merger (Renggli et al., 2022) (real score: 118, params: $\sim$77M) which incorporates a lightweight module that merges redundant tokens between transformer layers, both performed well under limited training epochs (25), indicating strong inductive biases and fast convergence capabilities. Their performance suggests that such design choices may significantly enhance model efficiency and should be considered in future ViT developments for medical imaging tasks.

All models, except for Token-to-Token ViT, were successfully trained under the same computational environment. Most models converged well within 25 to 50 epochs, with higher epoch budgets yielding more stable learning curves (e.g., LeViT with 100 epochs). Models trained with fewer epochs (like CaiT and xcit) exhibited more variability in performance, reinforcing the need for longer training schedules especially for larger architectures.

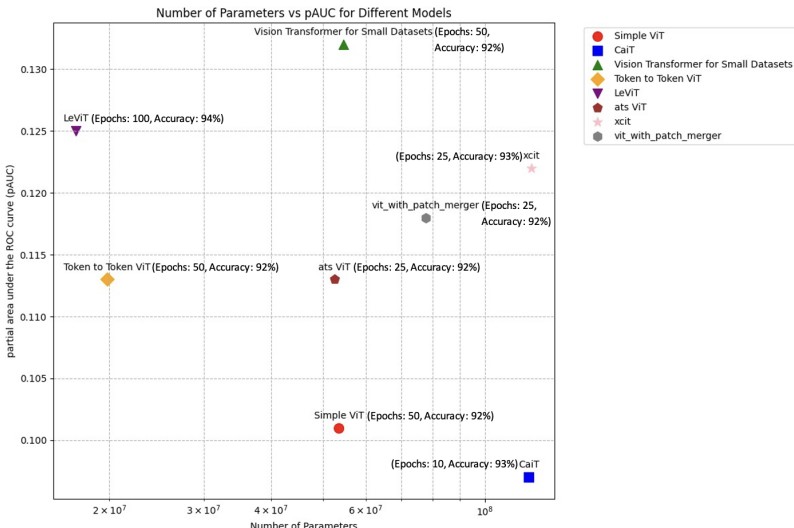

Figure 1: Models performance namely real score were described as partial area under the ROC curve (pAUC) above 80% true positive rate (TPR) since the TPR below 80% is unacceptable in clinical practice.

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
