# OpenReview forum: "Early diagnosis of skin cancer from phone-taken skin lesion images using Vision Transformers"
_MIDL.io/2025/Short_Papers — MIDL 2025 - Short Papers_

### Official Review · Reviewer_945U · 2025-04-28

**Rating:** 3
**Confidence:** 5

**Summary:**

The authors benchmark five recent Vision-Transformer (ViT) families—Token-to-Token, CaiT, LeViT, ATS-ViT and XCiT—on the ISIC 2024 Kaggle “phone-quality” skin-cancer dataset (~401 k cropped lesions) to explore early-stage, consumer-image diagnosis. They compare accuracy, a “real score” (partial-AUC above 0.80 TPR) and model complexity. The ViT-Small-Datasets model reaches 92% accuracy and the highest real score of 132, whereas LeViT attains the best accuracy (94%) with just 17 M parameters. CaiT underperforms despite its 120M parameters, illustrating diminishing returns of scale without sufficient training budget

**Strengths:**

+ Topical relevance – tackles early, smartphone-level images rather than late-stage dermoscopy, a useful but under-studied niche.
+ Comprehensive breadth – compares diverse ViT design philosophies (patch re-encoding, adaptive token sampling, hybrid CNN-attention) under a unified training pipeline.
+ Compute awareness – reports parameter counts, GFLOPs and training-epoch budgets, highlighting the efficiency–performance trade-off important for point-of-care or embedded devices ​

**Weaknesses:**

1. Lack of CNN benchmark: The study benchmarks only Vision-Transformer variants. Convolutional networks (e.g., ResNet-50, EfficientNet-B4, DenseNet-121)—especially when pre-trained on ImageNet—are still the de-facto standard in medical-image classification, and they often remain competitive or superior on small datasets. Without at least one representative CNN trained and evaluated under the same protocol, the reader cannot gauge whether the reported ViT gains are meaningful or merely artifacts of the model family chosen.
2. No statistical analysis – results lack confidence intervals or significance testing, so whether 1-2 % accuracy gaps matter is unclear.
3. Figure 1 lacks error bars; font sizes are hard to read.

---

### Decision · Program_Chairs · 2025-05-01

Accept